# Towards a concrete landscape: Assessing the efficiency of land consumption in the Greater Accra Region, Ghana

**Adams Osman**[1]*, **David Oscar Yawson**[2], **Simon Mariwah**[3], **Ishmael Yaw Dadson**[1]

**1** Department of Geography Education, University of Education, Winneba, Ghana, **2** Center for Resource Management and Environmental Studies, The University of the West Indies, Wanstead, Barbados, **3** Department of Geography and Regional Planning, University of Cape Coast, Cape Coast, Ghana

* aosman@uew.edu.gh

## Abstract

Most existing studies on land consumption have used a reactive approach to assess the phenomenon. However, for evidence-based policies, an initiative-taking forecast has been touted to be more appropriate. This study, therefore, assessed current trends and efficiency of land consumption in the Greater Accra Region from 1987 to 2017, and predicted a 30-year future land consumption in a "business-as-usual" scenario. The study adopted maximum likelihood image classification techniques and "combinatorial or" to model land cover change for Greater Accra from 1987 to 2017 while the UN-Habitat land efficiency index was employed to model efficiency of land consumption. In addition, Leo-Breiman Forest based regression, was used to model a future land cover by using the 30 years land cover change as a dependant variable and a series of natural and anthropogenic factors as independent variables. Results showed that artificial surfaces increased from 4.2% to 33.1%, with an annual growth rate of 22.1% in 30 years. Land consumption was highly inefficient as only 4.2% of the region had a good proportion of population per land area. Factors which influenced artificial surface growth were population, distance from water bodies, poverty index, distance from sacred groves, proportion of agriculture population with a small margin of influence from soil and geology type. Landscape prediction showed that artificial surfaces will increase to 92.6% as more places are coated with concrete. The high rate of land inefficiency provides an opportunity for re-zoning by the Land Use and Spatial Planning Authority of Ghana to accommodate the growing population.

## Introduction

Human transformation and its impacts on the earth's biophysical environment are becoming more glaring and intense in the 21st century, with an estimated 37.5% and 1.3% of the world's land mass being agriculture landscape and urban landscape respectively [1]. Causes of land consumption include increasing population, industrialisation, economic growth, unplanned development, unsecured land tenure, governmental policies and food demand [2–5]. Between

**Data Availability Statement:** The data underlying the results presented in the study are available from the Figshare database (https://doi.org/10.6084/m9.figshare.19300100.v1).

**Funding:** The authors(s) received no specific funding for this work

**Competing interests:** The authors have declared that no competing interests exist

agriculture and urban areas, the latter has serious dents on the land because once instituted, it becomes irreversible to earlier natural ecology.

The non-renewability of urban landscapes is due to the concretisation of surfaces which causes land sealing; thus, preventing the growth of flora and fauna [6, 7]. It also affects hydrological cycles as it disrupts percolation and groundwater recharge, thereby increasing the incidence of flooding [8]. It also leads to wetland loss, sequestration of carbon and loss of biodiversity due to high irreplaceable fauna and flora populations [9]. Urban land consumption is associated with long commuting distances and time, heat island effect, and air pollution [10]. Also, the increase in urban concrete surfaces contributes to increasing psychological stress because urban folks become detached from nature [11]. The negative effects of land consumption make it undesirable and a threat to the achievement of a sustainable world as captured in the Sustainable Development Goal 11. It is, thus, essential that patterns of land consumption are readily available to planners for strategic land use planning.

Although predicting future land consumption has a great advantage for streamlining development in less developed areas, further analysis in spatial land efficiency helps to support planning in already land-consumed areas [12]. Land efficiency is the measure of a unit area of physically developed land as a function of socio-economic activities [13]. Factors influencing land efficiency are economic growth, foreign direct investment, institutional capacity, government policy, productivity, consumption, labour, property rights and urbanisation [12, 14]. Measuring land efficiency provides a means to ascertain if land consumption is of ecological, environmental or social good. Also, land efficiency assessment serves as a guide to measure population growth and land availability and forge proper planning schemes [3]. One weakness of land efficiency is that land consumption at the initial stage produces efficient agglomeration but, but without planning, it later transgresses to agglomeration diseconomies [4]. Future diseconomies of land consumption can be tackled through land use planning systems supported by efficient land markets and policing [4, 5]. Policies such as setting urban boundaries and green zones with greater enforcement best control land consumption for efficiency [15].

Land consumption in Greater Accra Region has been on the increase with limited studies on the spatial growth of the entire region as existing studies mostly focus on the districts and watershed zones within the region [16–18]. Such haphazard growth can be attributed to the structural adjustment policy in Ghana which detached government from housing without providing a guided spatial planning policy [17]. Despite the acceptance of the rapid spatial growth of the region, efficiency of the growth is unknown, which inhibits spatial planning/re-demarcation of already built zones and outward planning of natural lands. Based on these gaps, the study proposes the following hypotheses:

H1: *There is no significant difference in artificial surfaces consumption and other dominant land cover types in Greater Accra Region from 1987 to 2017.*

H2. *Artificial land consumption in the region is not efficient.*

H3: *Artificial surface (concrete landscape) is less likely to consume all remaining terrestrial land covers in Greater Accra Region.*

This study aims to provide spatial information for planners by identifying areas with low land efficiency for rezoning and re-development to reverse trends of diseconomies of agglomeration of Greater Accra Region and the consequences of not reversing the current trends. Business-as-usual physical development in Greater Accra has the propensity to result in natural environmental disasters for ecosystems, with reciprocal effects on inhabitants of the region.

## Land change and consumption: Theoretical perspectives

Several hypotheses including natural, Malthusian, urban and regional economic growth and structural change explain land cover change and consumption. According to the natural theory, non-human forces were the most powerful drivers of land cover change before the anthropcene. Natural fires can damage vegetated land, converting it to bare land covering [19], whereas flooding leaves silt and sand deposits [20]. Volcanic eruptions, as well, have been observed to damage existing natural covers with sima [21] while elevation and soil factors can impact land cover types, since mountain locations with shallow soil might restrict vegetation growth [22]. On the other hand, the Malthusian hypothesis of land cover change explains that growing populations lead to the population outpacing food output, resulting in natural land consumption for farms and urban/built land cover [23].

Furthermore, population expansion drives the majority of agricultural households into off-farm jobs as a means of survival, freeing up space for built land cover. Also, market factors determine land consumption according to urban and regional economic theories [24]. They illustrate how land usage extends from the centre with industry, administrative buildings, and dwellings. Finally, the structural theory describes how interactions across institutional components (political, legal, administrative, economic and traditional) impact land consumption with policies at the national, regional or local levels having different effects. Market liberalisation, privatisation and currency depreciation are examples of structural adjustment policies that have the potential to increase growth and demand for land [25]. Furthermore, the central government's policies on lending, land rent, housing and subsidies to specific sectors of the economy may result in fast land consumption [26].

## Resource efficiency theory

Resource efficiency is an estimation of the output of a process as the ratio of achieved effect and the resources used to achieve the outcome [27]. The main tenet of resource efficiency theory is to gain more with less [28]; that is, using less resources such as land, water and energy to produce more of human desired goods and services. Resource efficiency advocates resource conservation and protection to reduce environmental impact and ensure sustainable development [28]. The land efficiency concept takes its roots from resource efficiency theory. Using land more efficiently involves using smaller areas of land to produce the same product or service [27]. The concept of land efficiency is based on the output or input-oriented perspectives [4]. The output-oriented perspective measures land efficiency as economic output per land unit [4]. Specifically, the output-oriented approach assesses the value added to land by secondary and tertiary industries per unit of square kilometer [4]. Output-oriented approaches ignore the social and environmental dimension of land consumption which are resolved by the input-output approaches [29].

Factors which influence land efficiency are land price, transportation networks, social services, public road facilities, industrial clusters and government policies [4, 14]. Ecologically, variables such as landscape diversity and fragmentation are essential in land efficiency measures [30]. An appropriate measure of land efficiency demands compositing of economic, social, environmental and even institutional indicators to generate a weighted measure [3, 4]. The essence is to provide a more comprehensive, unbiased and rigorous outcome.

However, the [31] supports a simple measure of land efficiency based on population growth in the absence of complex models and data availability. The measure seeks to supply data and output to guide development [31]. In countries, especially developing countries, where spatial data availability is a challenge, the measure is still relevant in assessing land efficiency with the growing population and the spatial growth of cities. The United Nations Habitat's simple

measure of land efficiency was adopted for our study because of the unavailability of spatial economic data to support complex spatial modelling. In total, the land efficiency model is a comparable indicator; thus, serving as an indicator for an achievable goal and an indicator of consumable resources.

## Methodology

### Study area

The Greater Accra Region is one of the 16 administrative regions of Ghana, and it is bounded to the west by the Central Region of Ghana, east and north by the Volta Region and Eastern Region respectively and south, by the Gulf of Guinea (Fig 1). Geographic coordinates show that the region lies within the bounding box of 6˚6'34.07" N and 0˚30'28.76" W to the North-East and 5˚28'25.76" N and 0˚37'28.21" W to the South-West. It covers about 370,390.6 hectares (ha) of land. The region houses the national capital of Ghana (Accra), making it highly influenced by formal government policies, economic pressures and urbanisation. The population of the region is about four million people, with a growth rate of 3.1% and about 75% of the populace being migrants [32].

On the local front, the region is the traditional abode for the Ga/Dangme ethnic group. The main drainage system in the region comprises the Korle, Kphesie, Sakumono and Songo lagoons among others. It also houses rivers like the Akonyador, Ado, Densu, Odaw, Kyekudor, Kyekubor, Nasakyir, Oludor and Sege which are mostly degraded or choked by physical development and refuse. These rivers flow on a geological bed of quartz, schist, fluvial and lacustrine sediments for wetland areas [33]. The dominant soils in the region are vertisols, solonetz, luvisols and acrisols [33].

The region falls within the Dry Equatorial Climatic Zone. It has an average rainfall amount of 787mm to 1,200mm yearly [34] while temperatures are often in the range of 22˚C to 32˚C with a mean of 26.5˚C. The vegetation has adapted to the low rainfall and high temperature with a coastal savannah vegetation cover dominated by shrubs, grass and a few mangrove patches. Greater Accra Region is the economic hub of Ghana; urban centres in the region are service-oriented while rural districts are primarily of agriculture production and fishing.

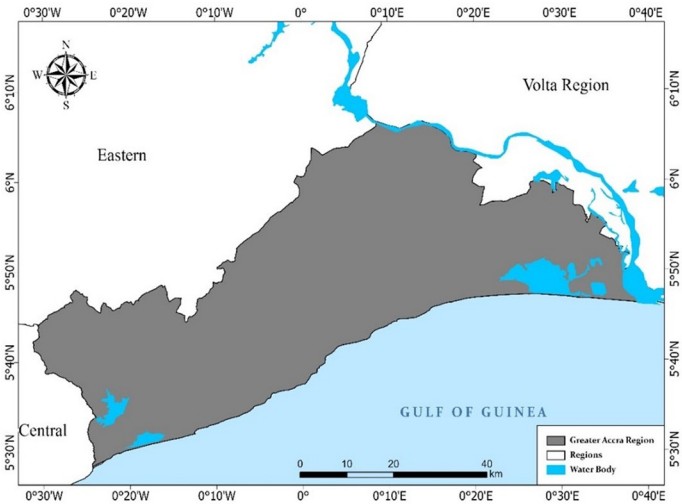

**Fig 1. Map of the Greater Accra Region of Ghana.** Source: [36].

## Data: Sources, processing and analysis

Data used for land cover mapping were the Landsat images of 1987, 2005 and 2017 for Greater Accra Region. The images were cleaned for haze and noise with the Erdas Imagine 2013 software. They were, further, projected from the UTM 30 North coordinated system to the Ghana Metre Grid Systems and clipped to the regional boundary of Greater Accra in Erdas Imagine 2013 software. Using the maximum likelihood classification method, and based on [35] land cover scheme (https://www.fao.org/3/X0596E/x0596e01f.htm#p381_40252), the study generated land covers such as Natural & Semi-Natural Terrestrial Vegetation [NSTV-OF] (areas where the vegetative cover is in balance with the biotope's abiotic and biotic forces. This describes vegetation which has not been planted by people but has been impacted by their activity and mainly of trees; Natural & Semi-Natural Terrestrial Vegetation [NSTV-SG] (areas where the vegetative cover is in balance with the biotope's abiotic and biotic forces. It consists of vegetation which has not been planted by people but has been impacted by their activities. It composes mainly of grass.); Natural & Semi-Natural Aquatic Vegetation [NSAV] (areas that are transitional between pure terrestrial and aquatic systems and where the water table, vegetation cover and floods are greatly impacted by water. For e.g., mangroves, marshes, swamps and aquatic beds); Natural & Semi-Natural Waterbodies [NSW] (areas that are naturally covered by water such as lakes and rivers); Artificial surfaces [AS] [areas with a man-made cover due to human activity such as construction (cities, towns, transportation), extraction (open mines and quarries), or waste disposal]; Cultivated and Managed Terrestrial Areas [CMTA] (areas where natural vegetation has been removed or changed and replaced with different forms of artificial vegetative cover, including crops planted for harvest) and, lastly, Bare Areas [BA] (areas that do not have an artificial cover as a result of human activities. They include bare rock areas and sands)

The 2017 land cover map had the highest accuracy in terms of producer, customer and Kappa. The least accuracy assessment result was recorded for the land cover map of 1987, as it had a customer accuracy of 72%, 68% for producer and 0.7 Kappa (Table 1). Therefore, a change-detection map was generated using the 1987 land cover map against the 2017 map.

Land consumption was calculated as $LCR = \frac{(LN(Urb_{(t+n)}/Urb_t)}{(y)}$ with y as years under study, $Urb_t$ as urban spatial extent in km$^2$ for past/initial year, Urb $_{(t+n)}$ as extent for the current year. Efficiency of land based on population growth rate (LCPC) was estimated as $LCPC = \frac{Urb}{P}$, where Urb = Built area and P = Population. Areas with population between 1–150 people per hectare are, thus, seen as inefficient, 151–250 as moderately efficient and 251 people and above per hectare as efficient [31].

The research considered two major factors of land cover change: natural and anthropogenic factors. The natural variables consisted of distance away from water bodies, elevation, geology, rainfall, slope, soil type and temperature (Fig 2). Distance from water bodies was generated as euclidean function of all river bodies in the region. Elevation was cropped from SRTM dataset from the [37] after it was processed for sinks and fills. Geology and soil type datasets were

**Table 1. Accuracy assessments of land cover classes.**

| Land cover (Year) | Customer accuracy (%) | Producer Accuracy | Kappa (K) |
|---|---|---|---|
| **1987** | **72** | **68** | **0.72** |
| 2005 | 84 | 70 | 0.68 |
| 2017 | 92 | 94 | 0.96 |

Source: Authors' construct, 2021.

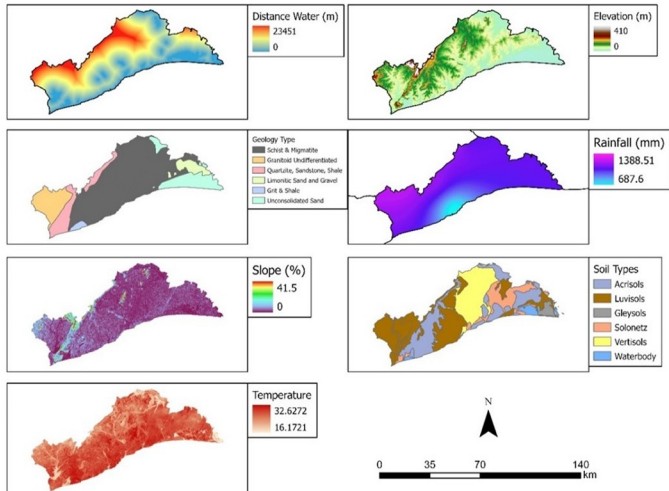

**Fig 2. Natural variables for land cover change in the Greater Accra Region.** Source: [36].

sourced from the Ghana Geological Service. Slope data was a derivative from the elevation data using the generate slope tool in ArcPro. Anthropogenic factors used in this study were proportion of agricultural population, distance from road, distance from sacred groves/land, land value, distance from market, distance from government residential facilities (housing), poverty index (population below poverty line) and population (Fig 3).

These factors were identified from the categorisation of causes of land cover change in Greater Accra by [17, 38, 39]. Also, the research adopted these causes because of their spatial measurability. Land prices were computed from a meta-analysis of land prices of lands in the Greater Accra Region as advertised by websites in Ghana. The average price of land advertised for a community was used and mapped spatially and later interpolated based on the moving average interpolation method. Distance from water bodies, road and nodal towns were all based on the Euclidean distance model in ArcPro. All datasets were clipped to fit the boundary of the study area and projected to the Ghana Metre Grid.

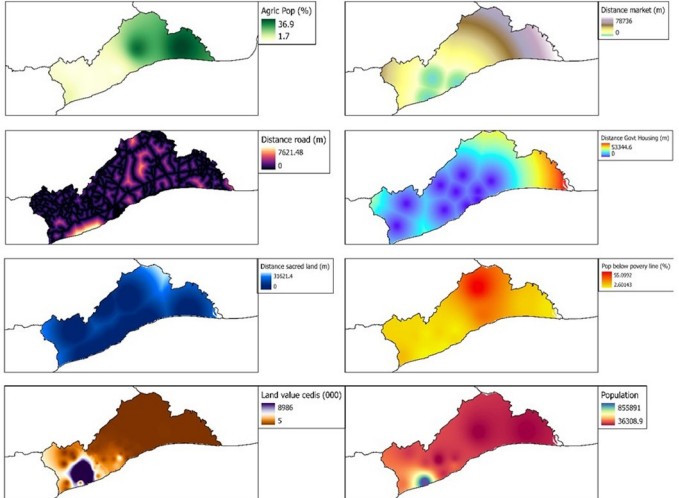

**Fig 3. Anthropogenic variables of land cover change in the Greater Accra Region.** Source: [36].

The study adopted a Maximum Variance Inflation Factor (VIF) cut-off of >7.5 to assess the multicollinearity of anthropogenic and natural factors of land cover change. Distance from the Accra Central Business District (CBD), Distance from government residential facilities and Distance from Market had VIF above the 7.5 threshold (Table 2). Distance from Accra CBD was dropped because of the high covariance with Distance from government residential facilities and Distance from Market.

Two variables (proportion of agriculture population and temperature) had a 100% importance significant value for the 4943 trails, with the least variable being Distance from Roads (78.11%) (Table 2). Five variables (distance from water bodies, proportion of agriculture population, temperature, distance from sacred groves and geology) produced the highest adjusted R-square value of 0.93 with Akaike's Information Criterion (-49037.44), Jarque-Bera p-value (0.00), Koenker (BP) Statistic p-value (0.00), Max Variance Inflation Factor (1.15) and minimum Spatial Autocorrelation p-value (0.1). Further, a Leo-Breiman Forest Based classification and regression (creates a model based on known values from a training dataset and later uses it to forecast unknown values with the same associated explanatory factors. It generates a large number of decision trees, known as forest, each tree creates its own forecast, which is then utilised as part of a voting mechanism to determine final predictions) [40] with the exploratory variables to predict between AS and CMTA (which is likely to consume the remaining terrestrial land cover in Greater Accra Region) were done.

Using the random forest regression, the future land consumption modelling generated 100 classification trees, 1 leaf size, 3 depth range of 4,083 to 4,404 and mean tree depth of 4,269. The model also had 3 randomly sampled variables, with about 30% of the dataset excluded for

**Table 2. Multicollinearity and importance of exploratory variables.**

| Variables | VIF | Covariates | VIF with Distance from Accra CBD | Sig% | Negative% | Positive% |
|---|---|---|---|---|---|---|
| Elevation | 2.68 | - | 2.66 | 93.20 | 19.24 | 80.76 |
| Distance from water | 2.47 | - | 2.47 | 92.59 | 29.57 | 70.43 |
| Population | 3.97 | - | 3.67 | 97.21 | 78.04 | 21.96 |
| Distance from Accra CBD | 27.11 | Distance from Market | - | | | |
| Distance from Roads | 1.16 | - | 1.16 | 78.11 | 50.85 | 49.15 |
| Proportion of Agricultural Population | 2.79 | - | 2.54 | 100 | 100 | - |
| Distance from Market | 14.54 | Distance from Accra CBD | 5.68 | 93.13 | 23.45 | 76.55 |
| Distance from Government Residential Facilities | 7.96 | Distance from Accra CBD | 4.23 | 92.52 | 48.20 | 51.80 |
| Rainfall | 2.26 | - | 2.26 | 99.86 | 4.49 | 95.51 |
| Poverty Index | 1.99 | - | 1.95 | 99.25 | 1.29 | 98.71 |
| Slope | 1.79 | - | 1.79 | 97.21 | 78.04 | 21.96 |
| Land Value | 3.49 | - | 3.49 | 99.66 | 79.27 | 20.73 |
| Temperature | 1.17 | - | 1.15 | 100 | - | 100 |
| Distance from Sacred Groves/land | 1.83 | - | 1.83 | 97.82 | 66.89 | 33.11 |
| Geology | 1.66 | - | 1.66 | 93.13 | 18.63 | 81.37 |
| Soil | 1.89 | - | 1.89 | 93.95 | 18.22 | 81.78 |
| Percentage of criterion passed | | Trails | No. passed | %passed | | |
| Min Adjusted R-Squared > 0.50 | | 4943 | 1471 | 29.76 | | |
| Max Coefficient p-value < 0.05 | | 4943 | 3949 | 79.89 | | |
| Max VIF Value < 7.50 | | 4943 | 4943 | 100.00 | | |
| Min Jarque-Bera p-value > 0.10 | | 4943 | - | - | | |
| Min Spatial Autocorrelation p-value > 0.10 | | 17 | - | - | | |

Source: Authors' construct, 2021.

validation after training the data with the independent variables' prediction for 30 years future land cover for Greater Accra Region. For the 30-year future land cover modelling, the study used the "business-as-usual" scenario, based on the assumption that the forces influencing land cover change are semi-permanent and result from all social and economic elements at play. Government has the greater influence to alter the trajectory of land cover change but with its less influence on housing and land ownership in the region since 1900's, such outcome is not expected [41, 42].

## Results

In 1987, the predominant land cover class was natural and semi-natural terrestrial vegetation (NSTV-SG). It covered an area of 161,628.9 ha (43.6%) (Table 3 and Fig 4). Cultivated managed and terrestrial areas (CMTA) was the second-largest cover for 1987 while natural & semi-natural waterbodies (NSW) had the least area of 3.41%. Natural & semi-natural terrestrial vegetation-shrubs and grass (NSTV-SG) remained the dominant land cover in 2005, but in 2017, artificial surfaces (AS) was the largest land cover with an area size of 122,650.0 ha (33.1%) (Table 3). AS land consumption from 1987 to 2005 was about 26,222.4 ha, giving an annual consumption rate of 9.07%. The level of AS consumption increased from 42,291.6 ha in 2005 to 122,650.0 ha in 2017. The annual growth rate of AS from 2005 to 2017 was 15.3%. But cumulative AS from 1987 to 2017 was 106,580.8 ha with an annual growth rate of 22.11%.

CMTA consumed a total of 37,468.9 ha of land area with an annual growth rate of 1.7% from 1987 to 2017.

The study extracted areas which have changed from other land covers to AS and CMTA for 1987 to 2005, 2005 to 2017 and 1987 to 2017 because they were the largest land cover types transforming the landscape of the study area. A test of independence was performed to assess whether there were significant differences between the means of areas changing to AS and CMTA. The study found statistically significant differences in the means of areas changing to AS (M = 3366.90m, SD = 28217.81) and CMTA (M = 6384.01m, SD = 24487.7); t (201097) = -3.46, p = 0.01 for 1987 to 2005 while for 1987 to 2017 in terms of differences in means of areas changing to AS (M = 19614.07m, SD = 76004) and CMTA (M = 7132.38m, SD = 214856.78); t (1355664) = -4.48, p = 0.00.

Based on the [31] land efficiency claification, the study identified three categories of land efficiency in Greater Accra Region (Fig 5). Efficiency values ranged from inefficient (1–150 people/hectare) to moderately efficient (151–250 people/hectare) and efficient (251 people and above/hectare). Efficient land use areas based on population were around the Accra Central

**Table 3. Landcover statistics of the Greater Accra Region for 1987 to 2017.**

| Land Cover | 1987 | | 2005 | | 2017 | |
|---|---|---|---|---|---|---|
| | Ha | % | Ha | % | Ha | % |
| AS | 16069.2 | 4.3 | 42291.6 | 11.4 | 122650.0 | 33.1 |
| BA | 13034.8 | 3.5 | 12677.9 | 3.4 | 17028.1 | 4.6 |
| CMTA | 72881.3 | 19.7 | 128480.4 | 34.7 | 110350.2 | 29.8 |
| NSAV | 26819.7 | 7.2 | 8423.3 | 2.3 | 18350.3 | 5.0 |
| NSTV-SG | 161628.9 | 43.6 | 164720.2 | 44.5 | 90169.7 | 24.3 |
| NSTV-OF | 67343.4 | 18.2 | 7520.8 | 2.0 | 2310.5 | 0.6 |
| NSW | 12613.3 | 3.4 | 6276.4 | 1.7 | 9531.8 | 2.6 |
| Total | 370390.6 | 100 | 370390.6 | 100 | 370390.6 | 100 |

Source: Authors' construct, 2021.

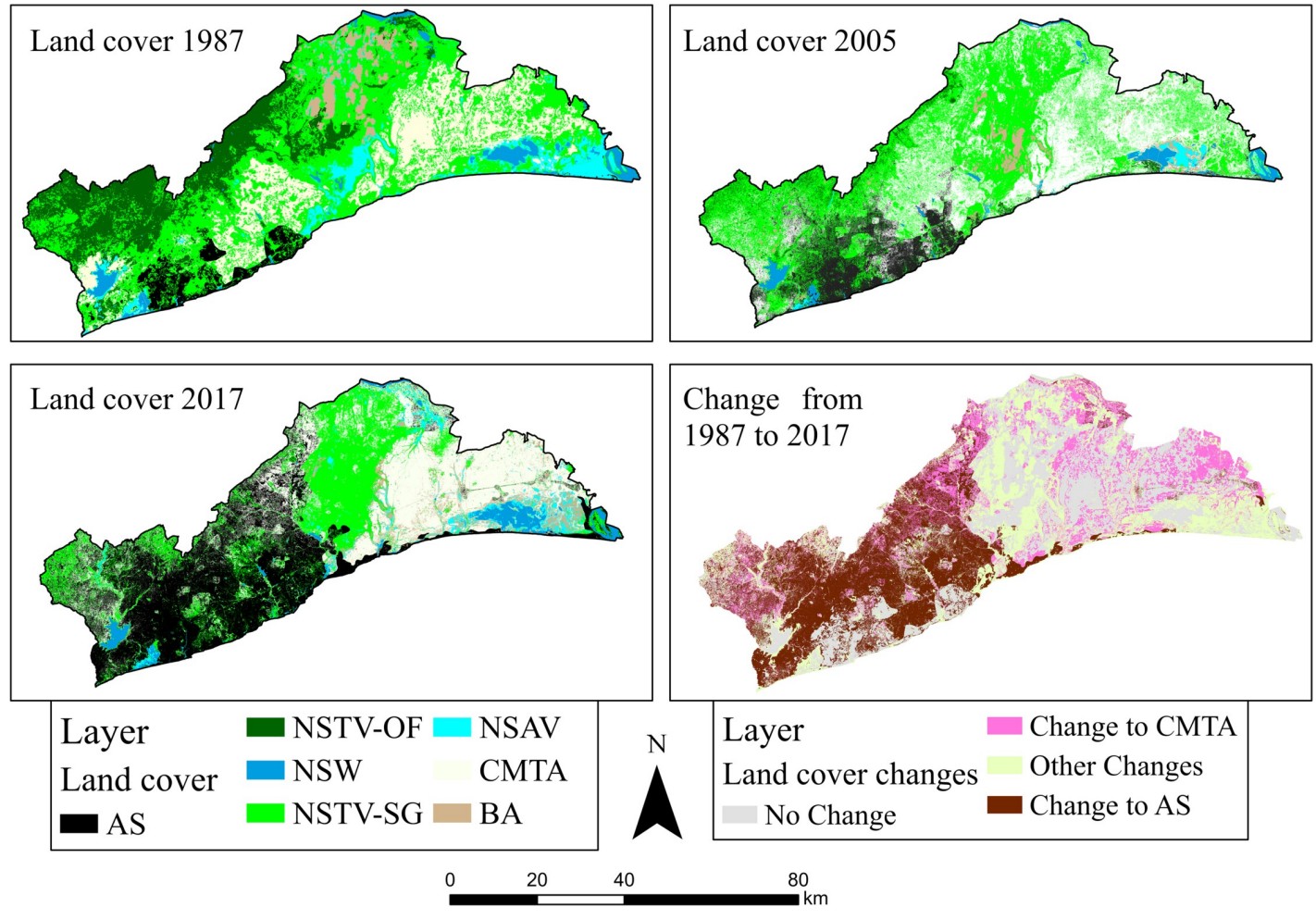

**Fig 4. Land cover and change maps for the study area for 1987, 2005 and 2017.** Source: Authors' construct with base map from [37].

and Gbawe enclave covering about 13,258.46 ha; that is, 10.81% of the entire AS in the Greater Accra Region.

Inefficient zone represented about 64.76% of the AS with the remaining 24.43% as moderately efficient areas. AS areas from Teshie to Ada Foah were in inefficient land use zones.

### Future land consumption of the Greater Accra Region (30 years from 2017)

The factors used for the Leo-Breiman forest based classification and regression model generated various levels of predictive importance, with population having the highest level of importance, predicting land cover change by 21.19%. The least important variable was temperature (0.0%) (Fig 6).

The model did check for validation of the training data by matching the results with 30% of the data not included in the training sample. The results for the validation data generated an F-Score of 0.92 and 0.95 for both AS and CMTA respectively. The sensitivity value reduced by a small margin to 0.95 and 0.93 for AS and CMTA and a Matthew Correlation Coefficient (MCC) value of 0.95 for both AS and CMTA. However, the explanatory variables had high validation shares within the range of 0.99 to 1 (Table 4).

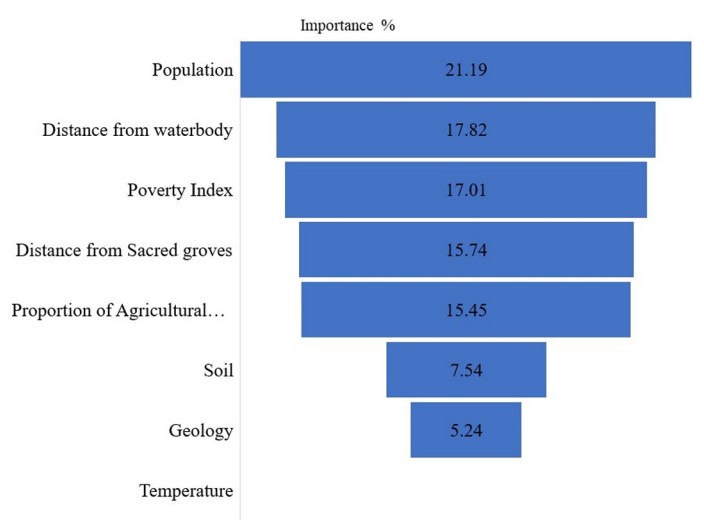

**Fig 5. Land consumption in Greater Accra Region.** Source: Authors construct with base map from [37].

**Fig 6. Level of importance for the variables.** Source: Authors' construct, 2021.

**Table 4. Accuracy of prediction of training and validation for the exploratory variables.**

| Variables | Training | | Validation | | Training Share | Validation Share |
|---|---|---|---|---|---|---|
| | Min | Max | Min | Max | | |
| Distance from waterbodies | 5.03 | 23387.72 | 14.51 | 23197.39 | 1 | 0.99 |
| Population | 0.00 | 1837290.13 | 0.00 | 1838587.88 | 1 | 1 |
| Proportion of Agricultural Population | 0.00 | 36.00 | 0.00 | 36.00 | 1 | 1 |
| Poverty Index | 2 | 55.00 | 2.00 | 55.00 | 1 | 1 |
| Temperature | 0.00 | 30.37 | 0.00 | 30.07 | 1 | 0.99 |
| Distance from Sacred Groves | 35.50 | 31349.59 | 51.03 | 31271.64 | 1 | 1 |
| Geology | 1 | 7.00 | 1 | 7.00 | 1 | 1 |
| Soil | 1 | 9 | 1 | 9 | 1 | 1 |

Source: Authors construct, 2021.

Per the Leo-Breiman Forest based classification and regression, AS and CMTA will be the only terrestrial land covers in the future (Fig 7). Of the remaining 238,208.8ha of terrestrial lands remaining in the region, about 220,569.8ha will be consumed by AS. This is going to take the entire land cover for AS in the region to an area size of 343,219.8ha (92.6% of the region). CMTA will reduce in size from 110,350.2ha to occupy an area of 17,639ha (about 84.01% of its original size). AS will consume all available terrestrial landscape into a homogenous concrete landscape.

## Discussion

The main purpose of this study was to assess the efficiency of land consumption based on the population of the Greater Accra Region using classified Landsat imagery for over 30 years. The natural land cover classes NSAV, NSTV-SG and NSTV-OF were dominant from 1987 to 2005, covering more than two-thirds of the region. The dominant natural land cover classes in 1987 reduced as artificial surfaces gradually increased in 2005 to become the largest single land cover in 2017. A study by the [37] showed that Greater Accra Region was the only region in Ghana

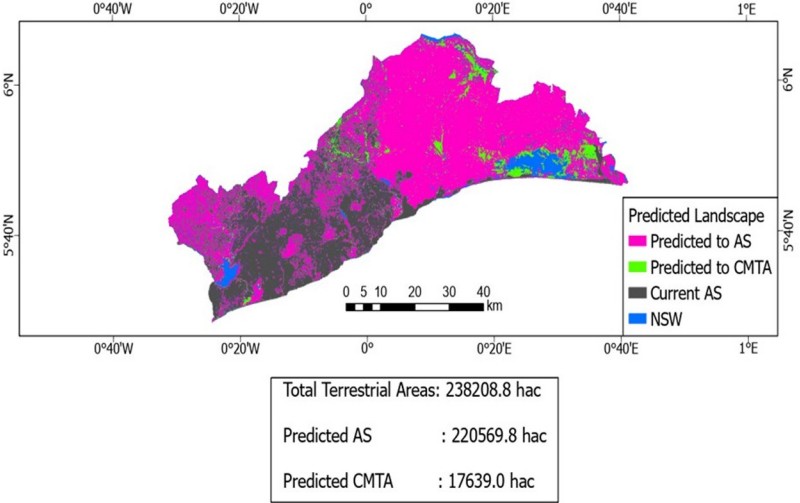

**Fig 7. Future land cover for Greater Accra Region in business-as-usual scenario.** Source: Authors' construct with base map from [37].

with highest growth of artificial surfaces as against agricultural landscape. This has rendered the land consumption in the region highly inefficient. Meanwhile, it has been established that high rate of artificial surface land consumption is a normal trend in developing cities across the world such as Beijing (China), Mumbai (India), Cairo (Egypt), and Lagos (Nigeria) [43–45].

[39] Population growth is the prime cause of the changing land cover in the Greater Accra Region [39]. This was confirmed by the random forest regression results in our current study. The findings were also in agreement with those of [3] in China [46], in Europe [47], in USA and [48] in South Africa all of whom declared population growth as the main determinant of land consumption in the world. In the Greater Accra Region, the increasing population is a result of in-migration rather than natural growth [49]. The pull factor of the region as the preferred destination for migrants is related to colonial and post-colonial government biased policies towards the region. Colonial rulers adopted Accra as their seat of governance; hence, provided the areas with better transport systems and educational, health, political and commercial infrastructure than other regions. Independent Ghana did not move from the spatially biased development of colonial leaders, but rather entrenched it with upgrading existing infrastructure in the region to the detriment of other regions [50, 51].

Although population is a major determinant of artificial surface growth in Greater Accra Region [52], claimed that structural policies in Ghana and the region also have a greater effect on the efficiency of land consumption. The structural adjustment programme adopted by Ghana in the 1980s led to the neglect of housing by government, a gap filled by low-income housing [52]. Also, loosely implemented land and spatial planning policies worsened the housing deficit, leading to a horizontal and non-uniform growth in the region. Weak institutions like the Land Use and Spatial Planning Authority, Forestry Commission and Environmental Protection Agency had little control over spatial growth [53] as artificial surfaces engulfed wetlands, groves and forest areas [18]. A well planned and executed spatial plan for Greater Accra could have reduced the large areas of land inefficiency in relation to population growth.

In this regard, [15] advocate the implementation of green belts to curtail land consumption while [54] advocates tradable land certificates to ensure land efficiency. Tradable land certificates are yet to be implemented in the study region and Ghana in general even though it is highlighted in the Land Policy of Ghana [55]. Also, the land management system in Greater Accra is traditionally communal; the government has limited direct control to enforce land trading and bonds. As a result [4], assert that in areas where governments have less control over land tenure systems, land consumption is high; hence, efficiency is low. With the absence of direct land control, an alternative is physically restricting horizontal growth while enforcing building codes and re-zoning and re-demarcating inefficient zones in the region [3].

With current trends of land consumption and land use inefficiency, firm action is needed to restrict the horizontal growth of the region into a homogenous concrete surface. Planners must focus on re-zoning inefficient land-use areas to improve the number of people per hectare as that will prevent rapid horizontal growth and encourage intensification. Per the forest-based modelling, it was seen that if horizontal growth is not curtailed, Greater Accra will develop into a homogenous concrete surface. This has serious physical, environmental and health effects in the region as espoused by [3] and [4]. That, in effect, will worsen current levels of urban heat [56], air quality [57], psychological health [58] and the dwindling small mammal population [59].

## Conclusions and recommendations

Urban land is a small fraction of natural lands degraded by humans compared to agricultural lands. Despite its limited spatial extent, urban growth leads to permanent concrete sealing of

land, preventing ecological growth within, on and around it. The effects of urban land consumption are even worsened with cyclical effects on human societies as it reduces resilience to climate change, flooding, urban heats and associated deaths and health complications. This study sought to assess the efficiency of urban land consumption in Greater Accra Region using remotely sensed datasets at three points in a 30-year interval. The study further modelled future land consumptions in the region based on a "Business-as-Usual" growth of urban artificial surfaces. Results showed that growth in urban land consumption was significantly higher than the growth of any other land cover in the Greater Accra Region. As result, land consumption in region was found to be moderately efficient but largely inefficient. This suggests a need for policy and management actions to constrain horizontal growth. The results point to potential areas for re-demarcation and re-zoning to improve efficiency. The scale of potential expansion of artificial surfaces in the future (based on the current trends) is huge and the consequences are alarming. It is extremely important, if not urgent, to adopt measures to reduce the rate of artificial surface expansion and coverage in the interest of sustainable development, ecosystem services and human wellbeing. The human and environmental costs of allowing the concretisation of the region will be far greater for government and society. Going forward, green infrastructure should be integral to physical development planning in the region.

## Author Contributions

**Conceptualization:** Adams Osman, David Oscar Yawson, Simon Mariwah.

**Data curation:** Adams Osman, David Oscar Yawson.

**Formal analysis:** Adams Osman.

**Methodology:** Adams Osman, Ishmael Yaw Dadson.

**Supervision:** Ishmael Yaw Dadson.

**Visualization:** David Oscar Yawson, Simon Mariwah.

**Writing – original draft:** Adams Osman, Simon Mariwah.

**Writing – review & editing:** David Oscar Yawson, Simon Mariwah, Ishmael Yaw Dadson.

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
