## [Decision Letter · Decision Letter 0]

4 Dec 2021

PONE-D-21-33010TOWARDS A CONCRETE LANDSCAPE? ASSESSMENT OF EFFICIENCY OF LAND CONSUMPTION IN GREATER ACCRA REGION, GHANAPLOS ONE

Dear Dr. Osman,

Thank you for submitting your manuscript to PLOS ONE. After careful consideration, we feel that it has merit but does not fully meet PLOS ONE’s publication criteria as it currently stands. Therefore, we invite you to submit a revised version of the manuscript that addresses the points raised during the review process.

Specifically, Reviewer 1 is against the publication of the manuscript on the grounds that it does not have merits on the contribution of literature both methodologically and scientifically. Reviewer2 suggests some minor issues need to be revised while Reviewer3 offers major revisions. I give major revision decision on this occasion however you should keep in mind that following the revisions, you have to persuade the reviewers on the suitability of your article to be published in the Journal. 

We look forward to receiving your revised manuscript.

Kind regards,

Eda Ustaoglu, PhD

Academic Editor

PLOS ONE

Journal Requirements:

"This research was possible through the financial assistance of the DAAD Performing Sustainability, Cultures and Development in West Africa."

"The authors(s) received no specific funding for this work"

4. We note that Figures 1 to 4 in your submission contain [map/satellite] images which may be copyrighted. All PLOS content is published under the Creative Commons Attribution License (CC BY 4.0), which means that the manuscript, images, and Supporting Information files will be freely available online, and any third party is permitted to access, download, copy, distribute, and use these materials in any way, even commercially, with proper attribution. For these reasons, we cannot publish previously copyrighted maps or satellite images created using proprietary data, such as Google software (Google Maps, Street View, and Earth). For more information, see our copyright guidelines: http://journals.plos.org/plosone/s/licenses-and-copyright.

a. You may seek permission from the original copyright holder of Figures 1 to 4 to publish the content specifically under the CC BY 4.0 license.  

Reviewers' comments:

Reviewer's Responses to Questions

**Comments to the Author**

1. Is the manuscript technically sound, and do the data support the conclusions?

Reviewer #1: Partly

Reviewer #2: Yes

Reviewer #3: Yes

2. Has the statistical analysis been performed appropriately and rigorously? 

Reviewer #1: No

Reviewer #2: Yes

Reviewer #3: I Don't Know

3. Have the authors made all data underlying the findings in their manuscript fully available?

Reviewer #1: Yes

Reviewer #2: Yes

Reviewer #3: No

4. Is the manuscript presented in an intelligible fashion and written in standard English?

Reviewer #1: Yes

Reviewer #2: Yes

Reviewer #3: Yes

5. Review Comments to the Author

Reviewer #1: This paper raised a concern of low land consumption efficiency in Greater Accra Region, by mapping the land cover change in 30 years, the authors calculated the land consumption efficiency using UN habitat’s LCPC and found that the land consumption in the region was highly inefficient. By running random forest model, the authors also identified variables that are associated with land consumption and predicted the future land cover change. The paper has solid sections in background and data, but relative weak in terms of the method and result interpretations.

Overall, the paper may have significant implications for the urban development in Greater Accra Region, especially how to balance the development of urban and population growth. However, I would say that neither the method nor the concluding remarks offer significant contribution to the existing literature, neither in theoretical nor practical domain. For international readers, we would like some take-away from the paper, but I failed to do so. I would suggest put more focus on the interpretation of the associations between land consumption and the influencing factors, and what can we learn from the results. Also, I am not convinced by the prediction results from the random forest model, as the method and model are not well introduced, and the results still require more in-depth interpretation.

Some other questions or comments are as follow:

Line 174: maybe give some brief introduction to each term, such as what is cultivated and managed terrestrial areas, or just provide a link to the FAO documents.

Line 180, I don’t quite understand how you decided to use the 1987 and 2017 maps to detect land cover change, as you said the accuracy of 1987 is the weakest.

Line 184: the efficiency of land is calculated here as the ratio between two growth rate, but why later was defined as population density (land consumption per capita)? Such as 1-150 people per hectare as inefficient. Also, I did not see the results of the LCRPGR in the rest of the paper, but only the LCPC.

Line 204: although you listed the references that support the selection of independent variables, I suggest add some explanations, especially the theoretical underpinnings behind the potential associations.

Line 237: should be Table 3.

Line 247, use cumulative “AS”, and the number should be 106580.8

Reformate Table 2 and Table 4 to fit the pages.

Line 278: the random forest modelling should be provided with more details. I am not an expert on random forest model and correct me if I am wrong. I understand that you were using the random forest to regress the correlations between land cover change and the influencing factors, and then using such correlations to predict the future change of land cover under “business as usual” scenario, however, more information should be provided, such as what is the target year of your prediction, have you set any thresholds of the influencing factors, such as the population growth (such as the world population projected by UN), climate change (maybe also from UN’s climate report?).

Line 283: the interpretation of the Fig 5, while I can tell from the Figure that population has the highest importance, but the text said poverty index has the highest level of importance.

I have read paper using population projection to predict land use change, or verse vise, but why do you think a random forest that includes all factors in the model is better?

Line 303: it should be “Leo-Breiman”.

Line 342: the authors pointed out the planning and protection policies intensified the land consumption, but how to include such influences in the predicting model?

What is the unit of analysis, especially when calculating the land efficiency? grid cells?

Actually, you were using the land consumption per capita (LCPC) from UN Habitat (https://unhabitat.org/sites/default/files/2021/08/indicator_11.3.1_training_module_land_use_efficiency.pdf) to calculate the “land use efficiency”, rather than the LCRPGR you mentioned (whether the increase of urbanized land is compatible with the increase of population, line 184). In addition, I did not find the threshold of low, moderately and efficient from the reference you provided (line 188, https://apo.org.au/sites/default/files/resource-files/2018-07/apo-nid182836.pdf).

Reviewer #2: This manuscript “TOWARDS A CONCRETE LANDSCAPE? ASSESSMENT OF EFFICIENCY OF LAND CONSUMPTION IN GREATER ACCRA REGION, GHANA” assess the efficiency of land consumption based on population in the Greater Accra Region using classified Landsat imagery over 30 years. Based on the prediction, Artificial surfaces and Cultivated and Managed Terrestrial Areas will be the dominant terrestrial land types in the future. The authors used the Landsat images of Greater Accra Region of 1987, 2005 and 2017 and Resource Efficiency Theory presented the study. Though the results were relatively simple, the contents and thoughts were a bit interesting. After some revision, I would recommend it for publication in PLOS ONE. Detailed revision listed below.

In the abstract part, studied methods and studied years (1987, 2005 and 2017) were not presented.

Many minor problems need to be revised, such as the “Li, Ye, Song, 2020;” in line 47, “Fig 1” in line 139, “(Yan, Peng & Wu, 2020).” in line 111, “(Ahmad, Pandey, & Kumar, 2019).” In line 53, “Guo and Xiong, (2014)” in 114, “Liu et al,” in line 120, “Grekousis and Mountrakis, (2015)” in line 327, in line 349, “Fig 1: Map …” in line 147 and in all the figure captions.

Please unify the tense you used. Such as “Stewart and Haaga (2018) assert that increasing” in line 56-57, and all the primary tenses when you cited and described other reports.

This manuscript had two “Fig. 3” in line 207 and in line 252, respectively, pleased revise them as well as in the text.

In line 282-285, “…predicting land cover change by 13.91% while the least important variables were rainfall (9.53%) and slope (6.60%) …” why the data “13.91%, 9.53% and 6.60%” was disagreed with them in Fig. 5?

Line 251, delete the second “organic”

Line 289, please revise “in at the experimental site”

Reviewer #3: The manuscript is well-developed and well-organized. The article is well-written with the analyses quite useful to identify land consumption of an area. I think it is suitable to be published in PLOS ONE.

I am looking forward to reading the revised paper after the following suggest edits.

* For which year land cover trends is simulated need to be precisely stated.

* Future trends of land consumption is evaluated using “business as usual scenario”. Why business as usual scenario was chosen for the analysis need to be properly justified. Given that it is unlikely to continue the existing growth rate for prolonged time. And, this model is susceptible for portraying an upper bound benchmark for urban growth.

* Few caption of table and tables were misplaced. Need to be corrected.

6. PLOS authors have the option to publish the peer review history of their article (what does this mean?). If published, this will include your full peer review and any attached files.

Reviewer #1: No

Reviewer #2: No

Reviewer #3: **Yes: **Farhan Asaf Abir

---

## [Author Response · Author response to Decision Letter 0]

1 Apr 2022

REVIEWER 1

Comment 1

Line 174: maybe give some brief introduction to each term, such as what is cultivated and managed terrestrial areas, or just provide a link to the FAO documents.

Response 1

Marked Line 208-226

Unmarked Line 195-210

Brief explanation is provided for each land cover type. Also link to assess in-depth meanings is provided

Comment 2

Line 180, I don’t quite understand how you decided to use the 1987 and 2017 maps to detect land cover change, as you said the accuracy of 1987 is the weakest.

Response 2

Satellite image for the area starts from 1987. The year 2017 was used because at the time of the study that was the current year and it also provided a 30 year span for the study. The 1987 had few bands compared with the 2005 and 2017 possibly accounting for less accuracy.

Comment 3

Line 184: the efficiency of land is calculated here as the ratio between two growth rate, but why later was defined as population density (land consumption per capita)? Such as 1-150 people per hectare as inefficient. Also, I did not see the results of the LCRPGR in the rest of the paper, but only the LCPC

Response 3

Marked Line 236

Unmarked Line 222

Function modified to Land consumption per capita mean built up area divided by population

Comment 4

Line 204: although you listed the references that support the selection of independent variables, I suggest add some explanations, especially the theoretical underpinnings behind the potential associations.

Response 4

Marked line 112

Unmarked line 103

Section on theory added

Heading: Land cover change and consumption

Comment 5

Line 237: should be Table 3

Response 5

Marked line 312

Unmarked 300

Corrected to Table 3

Comment 6

Line 247, use cumulative “AS”, and the number should be 106580.8

Response 6

Corrected

Comment 7

Reformate Table 2 and Table 4 to fit the pages

Response 7

Tables made to fit

Comment 8

Line 278: the random forest modelling should be provided with more details. I am not an expert on random forest model and correct me if I am wrong. I understand that you were using the random forest to regress the correlations between land cover change and the influencing factors, and then using such correlations to predict the future change of land cover under “business as usual” scenario, however, more information should be provided, such as what is the target year of your prediction, have you set any thresholds of the influencing factors, such as the population growth (such as the world population projected by UN), climate change (maybe also from UN’s climate report?).

Response 8

Marked line 281-284

A basic explanation is provided

[creates a model based on known values from a training dataset and later uses it to forecast unknown values with the same associated explanatory factors. It generates a large number of decision trees, known as forest, each tree creates its own forecast, which is then utilised as part of a voting mechanism to determine final predictions] (Bühlmann, 2010).

Marked line 287-296

Unmarked line 274-278

Using the random forest regression, the future land consumption modelling generated 100 classification trees, 1 leaf size, 3 depth range of 4083 to 4404, and mean tree depth of 4269. The model also had 3 randomly sampled variables, with about 30% of the dataset excluded for validation. After training the data with the independent variables, prediction for 30 years future land cover for Greater Accra Region. 

Comment 9

Line 283: the interpretation of the Fig 5, while I can tell from the Figure that population has the highest importance, but the text said poverty index has the highest level of importance.

Response 9

Marked Line 347

Unmarked line 333

Rewriting: The factors used for the model generated various levels of predictive importance, with population having the highest level of importance, predicting land cover change by 21.19% while the least important variables was temperature (0.0%) (Fig 6).

Comment 10

I have read paper using population projection to predict land use change, but why do you think a random forest that includes all factors in the model is better?

Response 10

The factors which influence land cover change are theoretically and empirically anthropogenic and natural. Most studies depend on anthropogenic by using population growth but there are other factors as the modelling indicated distance from water bodies, poverty and distance from sacred groves influence growth. These are micro level factors which needs to be accounted for.

Comment 11

Line 303: it should be “Leo-Breiman

Response 11

Correction made

Comment 12

Line 342: the authors pointed out the planning and protection policies intensified the land consumption, but how to include such influences in the predicting model?

Response 12

Modelling policies was difficult

Proxy variables were distance from road, distance from government housing

Comment 13

What is the unit of analysis, especially when calculating the land efficiency? grid cells?

Response 13

Hectare from 

https://archive.unescwa.org/sites/www.unescwa.org/files/u593/module_3_land_consumption_edited_23-03-2018.pdf

The grid cell was 30*30 meters because of the Landsat image used for classification

Comment 14

Actually, you were using the land consumption per capita (LCPC) from UN Habitat (https://unhabitat.org/sites/default/files/2021/08/indicator_11.3.1_training_module_land_use_efficiency.pdf) to calculate the “land use efficiency”, rather than the LCRPGR you mentioned (whether the increase of urbanized land is compatible with the increase of population, line 184). In addition, I did not find the threshold of low, moderately and efficient from the reference you provided (line 188, https://apo.org.au/sites/default/files/resource-files/2018-07/apo-nid182836.pdf).

Response 14

Mode to LCPC

Correction made in methodology

Interpretation for the levels of land efficiency can be found here

https://archive.unescwa.org/sites/www.unescwa.org/files/u593/module_3_land_consumption_edited_23-03-2018.pdf

REVIEWER 2

Comment 1

In the abstract part, studied methods and studied years (1987, 2005 and 2017) were not presented.

Response 1

The study adopted maximum likelihood image classification techniques and “combinatorial or “to model land cover change for Greater Accra from 1987 to 2017. While the UN-Habitat land efficiency index was employed to model land consumption and efficient of land cover change. Leo-Breiman Forest based regression was used to model future land cover by using the 30 years land cover change as dependant variable a series of natural and anthropogenic as independent variables.

Comment 2

Many minor problems need to be revised, such as the “Li, Ye, Song, 2020;” in line 47, “Fig 1” in line 139, “(Yan, Peng & Wu, 2020).” in line 111, “(Ahmad, Pandey, & Kumar, 2019).” In line 53, “Guo and Xiong, (2014)” in 114, “Liu et al,” in line 120, “Grekousis and Mountrakis, (2015)” in line 327, in line 349, “Fig 1: Map …” in line 147 and in all the figure captions.

Response 2

Corrected

Comment 3

Please unify the tense you used. Such as “Stewart and Haaga (2018) assert that increasing” in line 56-57, and all the primary tenses when you cited and described other reports.

Response 3

Well noted and corrected

Comment 4

This manuscript had two “Fig. 3” in line 207 and in line 252, respectively, pleased revise them as well as in the text.

Response 4

Tables and Figure numbers verified and corrected

Comment 5

In line 282-285, “…predicting land cover change by 13.91% while the least important variables were rainfall (9.53%) and slope (6.60%) …” why the data “13.91%, 9.53% and 6.60%” was disagreed with them in Fig. 5?

Response 5

The factors used for the Leo-Breiman forest based classification and regression model generated various levels of predictive importance, with population having the highest level of importance, predicting land cover change by 21.19% while the least important variables was temperature (0.0%) (Fig 6).

Comment 6

Line 251, delete the second “organic”

Response 6

No organic

Comment 7

Line 289, please revise “in at the experimental site”

Response 7

However, the explanatory variables had high validation shares within the range of 0.99 to 1 (Table 4).

REVIEWER 3

Comment 1

For which year land cover trends is simulated need to be precisely stated.

Response 1

Marked Line 201-202

Unmarked line 190-191

 Data used for land cover mapping were the Landsat images of 1987, 2005 and 2017 for Greater Accra Region

Comment 2

Future trends of land consumption is evaluated using “business as usual scenario”. Why business as usual scenario was chosen for the analysis need to be properly justified. Given that it is unlikely to continue the existing growth rate for prolonged time. And, this model is susceptible for portraying an upper bound benchmark for urban growth.

Response 2

Marked line 291-296

Unmarked line 278-283

For the 30-year future land cover modelling, the study used the "business as usual" scenario, assuming that the forces influencing land cover change are semi-permanent and the result of all social and economic elements at play. Government has the greater influence to alter the trajectory of land cover change but with less influence on housing and land ownership since 1900’s such outcome is not expected (Gough & Yankson, 2000; Quarcoopome, 1992). 

Comment 3

Few caption of table and tables were misplaced. Need to be corrected.

Response 3

Corrected

---

## [Decision Letter · Decision Letter 1]

16 May 2022

TOWARDS A CONCRETE LANDSCAPE: ASSESSMENT OF EFFICIENCY OF LAND CONSUMPTION IN GREATER ACCRA REGION, GHANA

PONE-D-21-33010R1

Dear Dr. Osman,

We’re pleased to inform you that your manuscript has been judged scientifically suitable for publication and will be formally accepted for publication once it meets all outstanding technical requirements.

Kind regards,

Eda Ustaoglu, PhD

Academic Editor

PLOS ONE

Additional Editor Comments (optional):

Reviewers' comments:

Reviewer's Responses to Questions

**Comments to the Author**

1. If the authors have adequately addressed your comments raised in a previous round of review and you feel that this manuscript is now acceptable for publication, you may indicate that here to bypass the “Comments to the Author” section, enter your conflict of interest statement in the “Confidential to Editor” section, and submit your "Accept" recommendation.

Reviewer #1: All comments have been addressed

2. Is the manuscript technically sound, and do the data support the conclusions?

Reviewer #1: Yes

3. Has the statistical analysis been performed appropriately and rigorously? 

Reviewer #1: Yes

4. Have the authors made all data underlying the findings in their manuscript fully available?

Reviewer #1: Yes

5. Is the manuscript presented in an intelligible fashion and written in standard English?

Reviewer #1: Yes

6. Review Comments to the Author

Reviewer #1: In this revised manuscript, I think all comments have been well addressed, and I have no further questions.

7. PLOS authors have the option to publish the peer review history of their article (what does this mean?). If published, this will include your full peer review and any attached files.

Reviewer #1: No

---

## [Editor Report · Acceptance letter]

27 May 2022

PONE-D-21-33010R1 

TOWARDS A CONCRETE LANDSCAPE: ASSESSING THE EFFICIENCY OF LAND CONSUMPTION IN THE GREATER ACCRA REGION, GHANA 

Dear Dr. Osman:

I'm pleased to inform you that your manuscript has been deemed suitable for publication in PLOS ONE. Congratulations! Your manuscript is now with our production department. 

Kind regards, 

on behalf of

Dr. Eda Ustaoglu 

Academic Editor

PLOS ONE